# Ptychography retrieval of fully polarized holograms from geometric-phase metasurfaces

Qinghua Song [1], Arthur Baroni [2], Rajath Sawant[1], Peinan Ni[1], Virginie Brandli[1], Sébastien Chenot[1], Stéphane Vézian [1], Benjamin Damilano[1], Philippe de Mierry[1], Samira Khadir[1], Patrick Ferrand [2] & Patrice Genevet [1]✉

Controlling light properties with diffractive planar elements requires full-polarization channels and accurate reconstruction of optical signal for real applications. Here, we present a general method that enables wavefront shaping with arbitrary output polarization by encoding both phase and polarization information into pixelated metasurfaces. We apply this concept to convert an input plane wave with linear polarization to a holographic image with arbitrary spatial output polarization. A vectorial ptychography technique is introduced for mapping the Jones matrix to monitor the reconstructed metasurface output field and to compute the full polarization properties of the vectorial far field patterns, confirming that pixelated interfaces can deflect vectorial images to desired directions for accurate targeting and wavefront shaping. Multiplexing pixelated deflectors that address different polarizations have been integrated into a shared aperture to display several arbitrary polarized images, leading to promising new applications in vector beam generation, full color display and augmented/virtual reality imaging.

[1] Université Cote d'Azur, CNRS, CRHEA, Rue Bernard Gregory, Sophia Antipolis, 06560 Valbonne, France. [2] Aix Marseille Université, CNRS, Centrale Marseille, Institut Fresnel, 13013 Marseille, France. ✉email: Patrice.Genevet@crhea.cnrs.fr

Widely used 3D stereoscopic displays, which rely on directional or selective projection of polarized images in left and right eyes, respectively, have long been considered as the next generation of displays. Unfortunately, their image reconstruction generally relied on additional bulky optical components such as filters or polarizers, which failed to impact the market for large public applications. Rapid developments in both virtual and augmented reality (VR/AR) are currently generating increasing interest, promising revolutionary new applications in three-dimensional (3D) image rendering and computer-generated virtual imaging. The deployment of these technologies to large public requires compact and lightweight VR/AR headsets capable of addressing image projection independently. In recent years, ultrathin and planar optical elements composed of subwavelength scale elements capable of encoding amplitude, phase and polarization information have been used to manipulate the wavefront of light[1–7], enabling holographic projection[8–10], thus offering new avenues for lightweight VR/AR headsets. Such artificial interfaces, also dubbed metasurfaces and meta-holograms are based on resonant structures[11,12] and/or geometry phase (or Pancharatnam–Berry (PB) phase)[13–15] to cover a full $2\pi$ phase range for encoding the arbitrary holographic phase profile. However, the polarization properties of the meta-hologram image are usually based on co- and cross-polarization related to the incident polarization[16,17], leading to limited polarization channels for real applications. Among the other attempts in achieving polarization-dependent holographic metasurfaces, we point out that reflective plasmonic metasurfaces utilizing two plasmonic nanorods per period with different distance and orientation angle can handle arbitrary polarization states but only efficient for oblique incidence[18]. Vectorial meta-holograms are also realized by combining both the propagation phase and PB phase, which suffers from narrow bandwidth[19,20].

Polarization reconstruction, which is based on the superposition of two orthogonal polarization bases, has been widely used in optical science. Circular polarization (CP), respectively, linear polarization (LP), can be generated based on the superposition of two orthogonal LP[21], respectively, CP[22]. However, full-polarization-reconstruction cannot be obtained based on such phase-only difference between two orthogonal bases and its application to arbitrary wavefront control have not been realized.

Here, we demonstrate a full-polarization-reconstructed metasurface that can produce arbitrary polarization for wavefront shaping based on a given LP incident light. The approach discussed in this paper relies on pixelated metasurfaces, in which each pixel acts as a deflector able to encode both the polarization and the holographic phase information, resulting in a holographic image in a specific angle with arbitrary polarization. In addition, the proposed method is broadband for all of the output linear polarizations. The experimental demonstrations, based on the metasurface Jones matrices extraction, are supported by state-of-the-art vectorial ptychography for full electromagnetic field characterization. By integrating multiple subpixels, a multidirectional meta-hologram is achieved for multiplexing polarization channels in different directions.

## Results

### Design method
The principle of the full-polarization-reconstruction is shown in Fig. 1a, consisting of two phase-gradient supercells. The polarization of the incident light is chosen to be LP in horizontal direction (LP-H) and is decomposed into two CP beams. Relying on PB phase information, any birefringent dielectric building block with an orientation angle of $\varphi$ can perform the conversion $|L\rangle \rightarrow e^{i2\varphi}|R\rangle$ and $|R\rangle \rightarrow e^{-i2\varphi}|L\rangle$, i.e., the LCP and RCP beams are transformed to opposite spin with a

PB phase of $2\varphi$ and $-2\varphi$, respectively. To address arbitrary polarization in each pixel, we compose the unit-cells in a counterclockwise-rotated bottom line of meta-molecules disposed with an angle increment of $\varphi_d$, i.e., to deflect the LCP light to the right side with a deflection angle of $\theta_t$, and a clock-wise rotated top line, arranged with the same angle increment of $\varphi_d$, to deflect the RCP light at the same deflection angle $\theta_t$. It should be noted that the output polarization from top and bottom lines to the interested deflected angle is pure RCP and LCP. Since the metasurface converts the handiness of the input CP beam and imposes a PB phase gradient in the converted CP beam, only the converted CP beam can be deflected to the interested angle and thus that pure CP beams from top and bottom supercells are obtained. Controlling the orientation angles of the bottom and top lines, i.e., choosing for example $\varphi$ and $-\varphi + \delta$, thus introducing independent transmitted phase, respectively, equal to $-2\varphi$ for the LCP and $-2\varphi + 2\delta$ for the RCP light, we introduce a phase difference of $-2\delta$ between the LCP and RCP light responsible for arbitrary polarization reconstruction. The dimension of the dielectric building block in the bottom and top lines can be also adjusted to further control the amplitude difference between RCP and LCP light. Thus, a new state of polarization (SoP) $|n\rangle$ can be refracted through the superposition of the two CP beams as

$$|n\rangle = a_R|R\rangle + a_L e^{-i2\delta}|L\rangle, \tag{1}$$

where $a_L$ and $a_R$ are the amplitude of the LCP and RCP beams, respectively. The azimuth angle $\psi$ and ellipticity angle $\chi$ of $|n\rangle$ are derived as $\psi = \delta$ and $\chi = \frac{1}{2}\arcsin\frac{a_R^2 - a_L^2}{a_R^2 + a_L^2}$. In this way, an arbitrary full polarization can be reconstructed by changing $\delta$ from $-\pi$ to $\pi$ and $a_L$, $a_R$ from 0 to 1, covering the entire Poincaré sphere as shown in Fig. 1b. The changing of $\delta$ leads to the shifting of the SoP along the latitude of the Poincaré sphere, while the changing of $a_R$ and $a_L$ makes the SoP shift along the longitude direction. When $\frac{a_R}{a_L}$ is increasing, the SoP moves toward the top of the Poincaré sphere and vice versa. It is worthy to note that introducing only a phase difference without amplitude modulation can only achieve LP without elliptical polarization (EP) (when $a_L = a_R$, $\chi$ becomes zero) (see Supplementary Fig. 1). The starting orientation angle of both lines in each pixel results in a detour phase that is exploited to address phase information of the meta-holograms (see Supplementary Figs. 2 and 3), which introduces another freedom to control the detour phase without shifting the structures by displacement of the aperture[23–25]. In order to multiplex the polarization of independent holographic images, we decomposed the metasurfaces in four subpixels (labeled as "C1", "C2", "C3", and "C4") with different orientation angle increment of $\varphi_{dn}$ ($n = 1, 2, 3, 4$ corresponded to C1, C2, C3, and C4, respectively, each responsible for the phase and polarization of an independent image at a given deflection angle). For design simplicity, we integrated these four subpixels into one super-pixel as shown in Fig. 1c, leading to a multi-directional meta-hologram with deflected angles given by

$$\theta_{tn} = \arcsin\left(\frac{2\varphi_{dn}}{k_0 p}\right), \tag{2}$$

where $k_0$ is the wavenumber in the free space and $p$ is the period of the unit-cell. The SoP of the four images can be independently designed to realize a full-polarization-reconstructed and multidirectional meta-hologram as shown in Fig. 1d.

### Full-polarized wavefront shaping
For the design of the meta-molecules, we performed full-wave finite-difference time-domain (FDTD) simulations at the wavelength of 600 nm. The unit-cell of the metasurface consists of GaN nano-pillars on a Sapphire

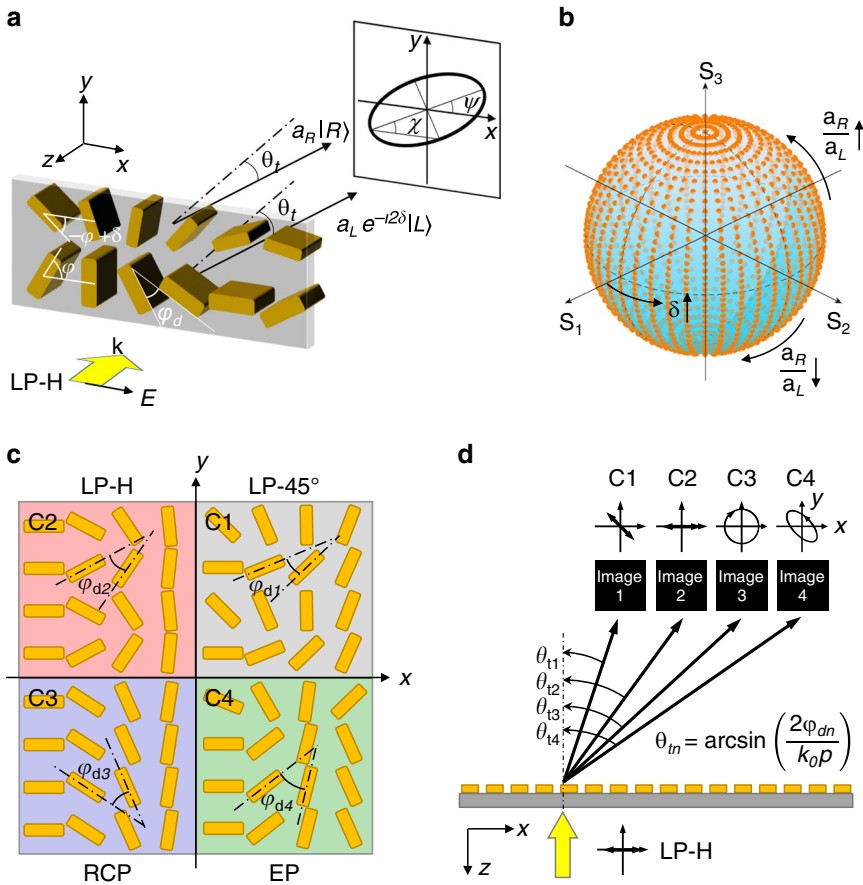

**Fig. 1 Full-polarization-reconstruction and multi-directional meta-hologram. a** Full-polarization-reconstruction through the superposition of two output CP beams with arbitrary phase and amplitude difference based on the same input LP beams. **b** Plotted Poincaré sphere covered full polarization with phase difference $\alpha$ from $-\pi$ to $\pi$ and amplitude $a_R$ and $a_L$ from 0 to 1. **c** Four subpixels of pixelated deflectors (labeled as C1, C2, C3, C4) representing to four SoP (LP-45°, LP-H, RCP, EP) are placed into one super-pixel to share the same aperture. **d** By encoding the holographic phase profile into the four subpixels array independently, multi-directional holographic images with different SoP are displayed in the direction of $\theta_{tn} = \arcsin\left(\frac{2\varphi_{dn}}{k_0 p}\right)$ ($n = 1, 2, 3, 4$).

substrate with a fixed dimension of $L_x = 230$ nm, $L_y = 120$ nm, $h = 1$ μm and various orientation angles $\varphi$ for the phase retardation as shown in the inset picture of Fig. 2a (see more details in Supplementary Figs. 5–7). The period of the meta-molecule is 300 nm, which is subwavelength to avoid unwanted diffraction grating, while remaining moderate to achieve fabrication. The incident LCP light is impinging on the pixelated metasurface from the backside, i.e., from the substrate to GaN nanopillar. Each GaN nanopillar acts as a waveplate that converts the spin of CP and introduces a PB phase. More than 91% conversion efficiency and $2\pi$ PB phase coverage can be realized in this configuration. Consequently, the metasurfaces with a footprint of 240 μm × 240 μm are fabricated using conventional nanofabrication process[26] as shown in Fig. 2b. In the experiment, a linear polarizer is used to generate LP-H incident light and a weakly focusing lens is employed to direct the light on the meta-hologram and produces the holographic image on the projector (see Supplementary Fig. 8). The optical behavior of the meta-hologram is exhaustively quantified and measured experimentally by means of optical vectorial ptychography[27,28]. This technique, which enables direct mapping of the metasurface Jones Matrices (see Methods and Supplementary Fig. 9), is applied to the characterization of vectorial metasurfaces. Relying on the monitoring of all exiting fields with a lateral sampling period smaller than 500 nm, including amplitude, phase and SoP properties, we could compute the far-field holographic image properties, and retrieve the entire map of the SoP of the generated

holographic fields for any given incident illumination polarization.

**Multi-directional and multiplexing meta-hologram**. As a proof of concept, four SoP of LP-45°, LP-H, RCP, and EP have been designed and experimentally demonstrated. Note that two different methods can be chosen to generate EP. The first one is using different size of GaN nano-pillars for LCP and RCP beams, which inevitably decreases the transmission efficiency. The second one consists in adjusting the number of rows for LCP and RCP beams, respectively, as exploited in this paper. Three rows for LCP and one row for RCP in a sub-pixel are demonstrated for the EP ($a_L = 3a_R$, $\chi = -26.57°$) and $\delta$ is chosen as $-22.5°$ ($\psi = \delta = -22.5°$). The orientation angle increment of the deflector is chosen as $\varphi_d = 45°$, leading to a deflection angle of $\theta_t = 30°$. The experimental results of the four beam deflectors agree well with the simulation results (see Supplementary Fig. 10). Subsequently, we encode the holographic phase profile using Gerchberg–Saxton (GS) algorithm into sub-pixel array for the realization of meta-hologram. Although the GS algorithm creates speckled images with random phase, it does not affect the SoP of the meta-hologram which is determined at the pixel level, instead of relying on the interference between different pixels. The designed structures for four SoP and the corresponding fabrication results are shown in Fig. 2c, d, respectively. Figure 2e shows that the holographic images are directed to 30° as expected. The measured SoP of the holographic images are consistent with the designed

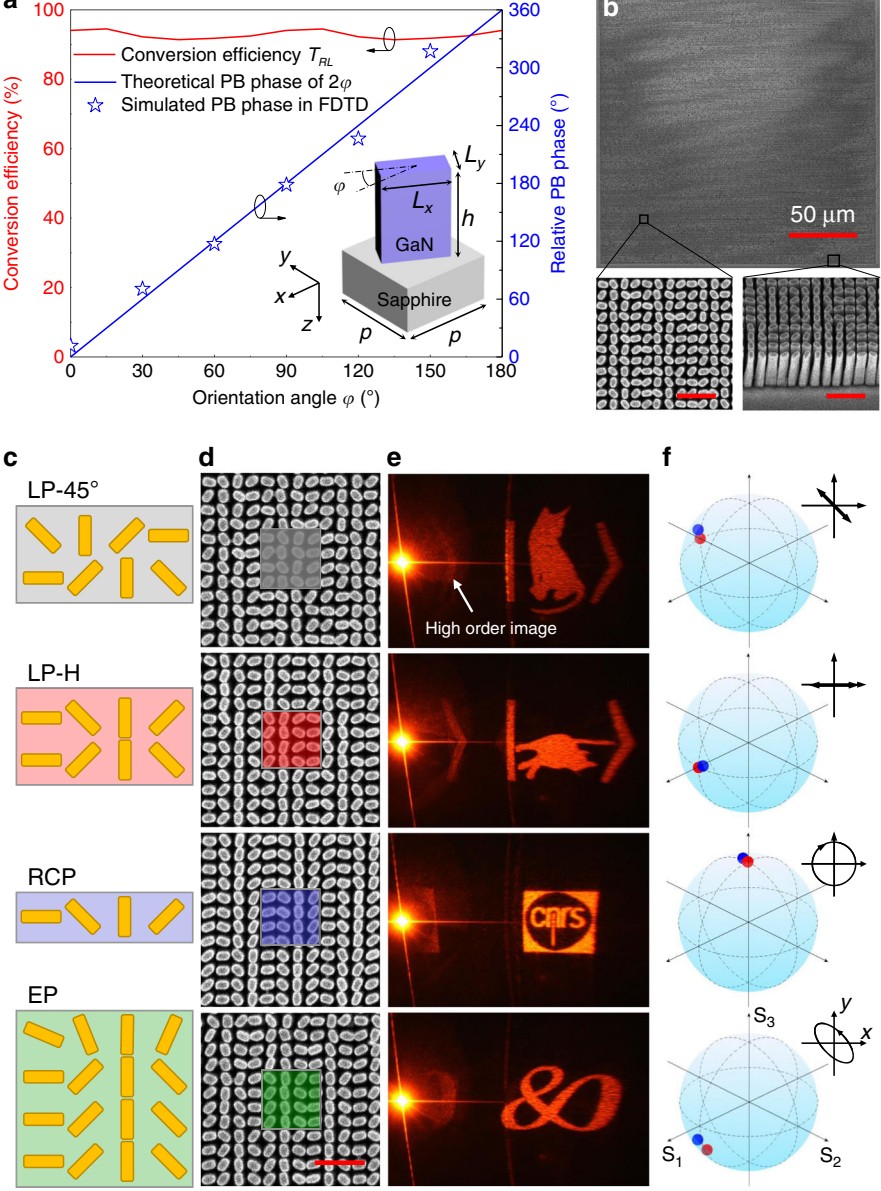

**Fig. 2 Realization of polarization-reconstructed meta-hologram. a** Simulated results of the CP conversion efficiency and relative PB phase of the metasurface consisting of an array of GaN nano-pillars fabricated on the top of a Sapphire substrate. The inset picture shows the schematic of one unit-cell of the metasurface with dimension of $L_x = 230$ nm, $L_y = 120$ nm, $h = 1$ µm, $p = 300$ nm and various orientation angle $\varphi$ from 0 to 180°. The incident LCP light is illuminated from the backside. **b** The scanning electron micrograph (SEM) images of a fabricated meta-hologram, where a top- and tilted-view of enlarged areas in the meta-hologram are presented. The scale bar of the enlarged image is 1 µm. **c** Structure configuration for the polarization generation of LP-45° (first row), LP-H (second row), RCP (third row), EP (last row). **d** SEM images of fabricated results. The scale bar is 1 µm. **e** Photographs of the meta-hologram images representing "alive Schrödinger's cat", "dead Schrödinger's cat", "CNRS logo 1", and "CNRS logo 2" with different SoP. High order image is induced near the central spot. **f** measured SoP of the corresponded images. The red and blue dots on the Poincaré sphere indicate the designed and measured polarization, respectively. Two Schrödinger's cats are adapted from Wikimedia.org.

SoP derived from Eq. (1) as shown in Fig. 2f. The absolute efficiency of each image at the interested order is $\eta_{LP+45°} = 17.2\%$, $\eta_{LP-H} = 15.3\%$, $\eta_{RCP} = 25.4\%$, and $\eta_{EP} = 20.3\%$, respectively. The drawback with this encoding pixelation method is that residual ghost images are produced due to the remaining polarization periodicity, equivalent to a grating effect according to (see Supplementary Fig. 12)

$$\beta_m = \arcsin\left(\sin\theta_t + \frac{2m\pi}{k_0 d}\right). \tag{3}$$

Indeed, grating effect occurs for uniformly distributed subpixel arrangement, producing multiple grating orders. To better comprehend the encoding method, we have proposed to demonstrate simple beam deflectors integrated with four subpixels arrays using two types of arrangement: uniformly or randomly distributed subpixels as shown in Fig. 3. The results show that for randomly distributed subpixels, i.e., when the lattice constant $d$ in Eq. (3) becomes variable, the grating effect is canceled. As a result, a pure desired order is obtained as shown in Fig. 3j. Four holographic phase profiles are then encoded into these four subpixels arrays denoted by "C1", "C2", "C3", and "C4" as shown in Fig. 4, and choosing four alphabet letters "C", "N", "R", and "S" we realized arbitrary polarization images representing the acronym of the "Centre National de la Recherche

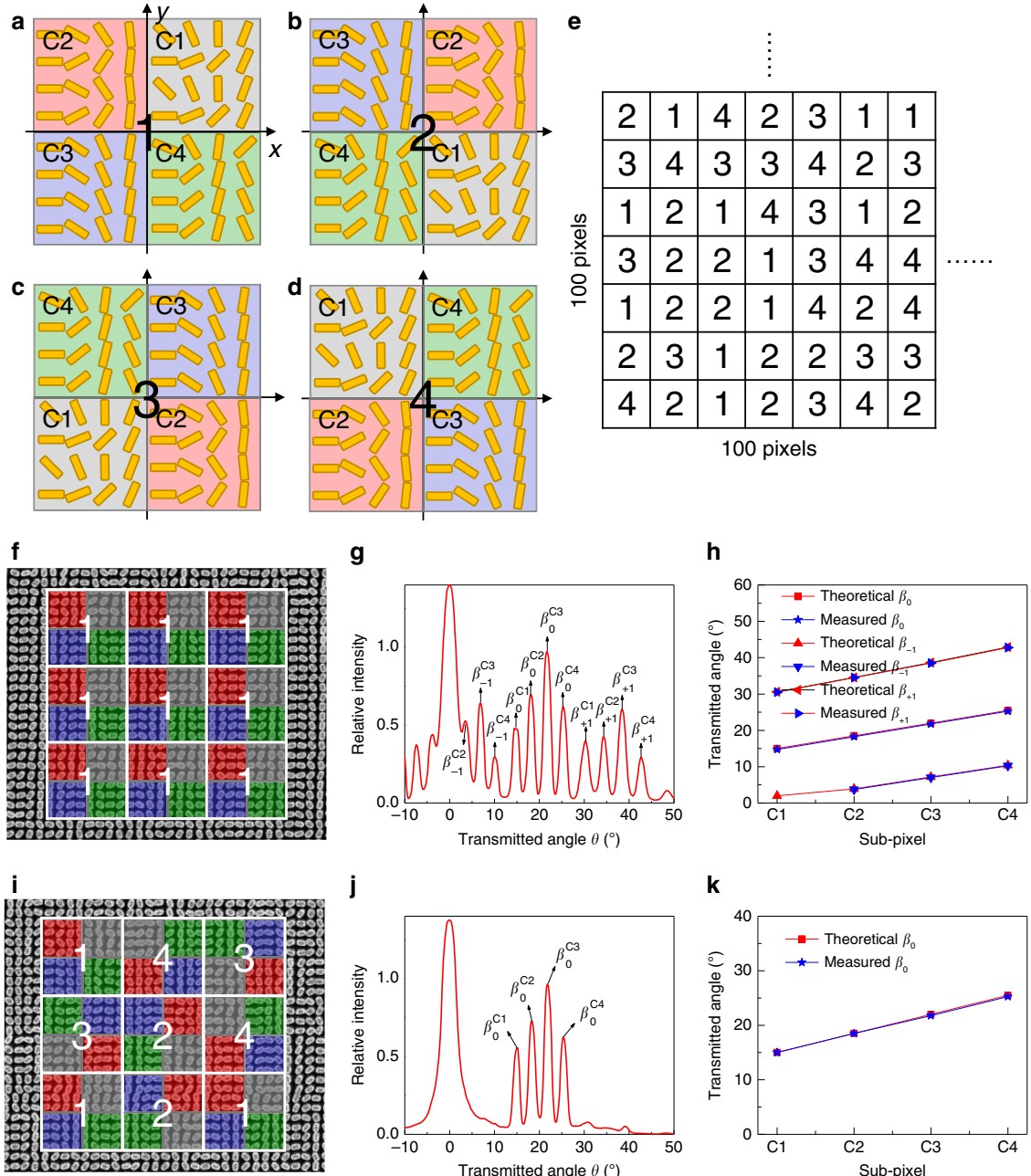

**Fig. 3 Uniformly and randomly distributed subpixels. a–d** The subpixels in one super-pixel are distributed with four different arrangement. **e** A 100 × 100 matrix with randomly distributed integer of 1, 2, 3, and 4 is generated using MATLAB. Each numeral is corresponding to the subpixels of (**a–d**). **f** A beam deflector is designed with subpixels uniformly distributed using configurations shown in (**a**). The four subpixels C1, C2, C3, and C4 are designed with interested order at 15°, 18.5°, 22°, and 25.5°, respectively. **g** Measured far-field intensity pattern shows that a series of grating orders $\beta_m^{Cn}$ ($m = \pm 1$, $n = 1$, 2, 3, 4) is generated. **h** Measured angles of the grating orders agree well to the simulated results. **i** A beam deflector design with randomly distributed subpixels based on the random matrix in (**e**). **j** Measured far-field intensity pattern shows the grating order is destroyed due to a variable lattice constant $d$. **k** Only interested order is obtained without other grating orders.

Scientifique" (see Fig. 1c). The orientation angle increments of the four pixelated deflectors are $\varphi_{d1} = 23.29°$, $\varphi_{d2} = 28.56°$, $\varphi_{d3} = 33.71°$, and $\varphi_{d4} = 38.75°$, which deflect the four hologram images to $\theta_{t1} = 15°$, $\theta_{t2} = 18.5°$, $\theta_{t0} = 22°$, and $\theta_{t4} = 25.5°$, respectively.

As expected, if the subpixels are uniformly distributed (see Fig. 4b), a series of ghost images produced by the grating effect are induced (see Fig. 4c) which can be eliminated with a random arrangement in Fig. 4d, thus exhibiting clean holographic images only at the desired orders as shown in Fig. 4e. The measured map of the Jones matrices for both periodic and random pixelated

interfaces have been realized in Fig. 4f, g (see more details in Supplementary Fig. 13). The corresponding far-field holographic image properties of the randomly distributed meta-hologram are shown in Fig. 4h, confirming remarkably the expected SoP orientations, and ellipticities for each letter of the pattern (see more details in Supplementary Fig. 14).

To further analyze the SoP of the four letters and confront vectorial ptychography with classical far-field measurements, we insert a polarimeter in front of each separated letter as shown in Fig. 5a. The measured SoP agrees well to the theoretical designs

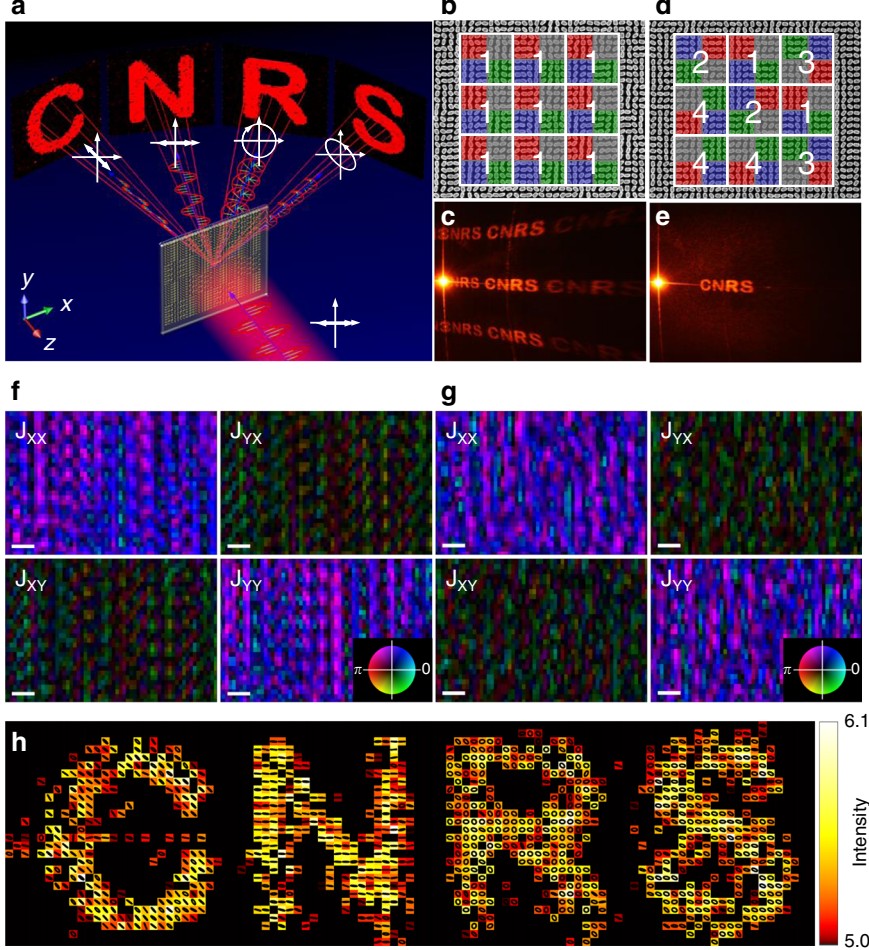

**Fig. 4 Realization of polarization-reconstructed meta-hologram with four sub-polarization pixels. a** Schematic of the polarization-reconstructed and multi-directional meta-hologram. **b** SEM image of the fabricated meta-hologram with uniformly distributed subpixels. **c** Photograph of the holographic image with a series of ghost images. **d** SEM image of the fabricated meta-hologram with randomly distributed subpixels. **e** Photograph of the holographic image with pure desired images. **f** Close-ups of Jones matrices maps of uniformly distributed and **g** randomly distributed meta-hologram, with phase encoded as hue and modulus encoded as brightness. Scale bars are 2.2 μm. **h** Intensity values (arbitrary units), together with the corresponding SoP, of the reconstructed vectorial far field of the metasurface.

and ptychography reconstructed fields (see Fig. 5b). A quarter waveplate with fast axis at angle $\theta_{\lambda/4}$ and a linear polarizer with transmission axis at angle $\theta_{LP}$ are selectively placed in front of the four images (see Fig. 5c and Supplementary Fig. 8) to block specific letters of "C", "N", "R", and "S" as shown in Fig. 5d, exhibiting multiplexing properties of the meta-hologram.

## Discussion

In conclusion, we have demonstrated a general method for wavefront shaping with arbitrary output polarization that allows full utilization of polarization channels. Pixelated deflectors are proposed to encode the information of polarization and phase profile. By randomly distributing the subpixels, a multi-directional meta-hologram is designed to display multi-holographic images in different directions. The general polarization imaging method discussed in this manuscript, could be applied for example to the realization of lightweight VR/AR displays. More importantly, such wavefront-shaping device that can spatially tailor the polarization and intensity light profile, could be further expanded toward a plethora of promising applications in vector beam generation, visible light communication, full color display, and animation production, etc.

## Methods

**Device fabrication.** The meta-holograms are fabricated by patterning a 1-μm-thick GaN thin-film grown on a double-side polished c-plan sapphire substrate using a molecular beam epitaxy (MBE) RIBER system. Conventional electron beam lithography (EBL) is used to pattern the GaN nano-pillars. A double layer of ~200 nm PMMA resist (495A4) is spin-coated on the GaN thin-film and then is baked on a hot plate with temperature of 125 °C. E-beam resist exposition is performed at 20 keV (Raith ElphyPlus, Zeiss Supra 40), followed by PMMA development using 3:1 IPA:MIBK solution. After Ni deposition of 50-nm thick using E-beam evaporation, the sample is immersed into the acetone solution for 2 h for the lift-off process, resulting in a Ni pattern as hard mask. Followed by reactive ion etching (RIE, Oxford system) with a plasma composed of $Cl_2CH_4Ar$ gases, the GaN nano-pillars are created. Finally, the Ni hard mask on the top of GaN nano-pillars is removed by using chemical etching with 1:2 $HCl:HNO_3$ solution. It is worth to note that there is possibility of realizing metasurface with similar functionalities by nanoimprint lithography using high dielectric constant and low loss materials. The proposed metasurface can be further applied to the curved or conformal substrates by using soft materials like PDMS[29–31].

**Optical setup.** The primary optical setup for characterizing the meta-hologram is shown in Supplementary Fig. 8. A laser beam at a wavelength of 600 nm propagates through a broadband linear polarizer (10GT04, Newport) with axis of transmission in horizontal direction. The LP-H beam is weakly focused by an achromatic lens with a focal length of 50 mm onto the meta-hologram, which is mounted on a three-dimensional translation stage. The holographic image is projected onto a projector placed 10 cm away from the meta-hologram. A selected quarter waveplate with fast axis at angle $\theta_{\lambda/4}$ and a linear polarizer with axis of transmission at angle $\theta_{LP}$ are used to block selected images. Assume an incident beam $\mathbf{E_{in}}$ propagates

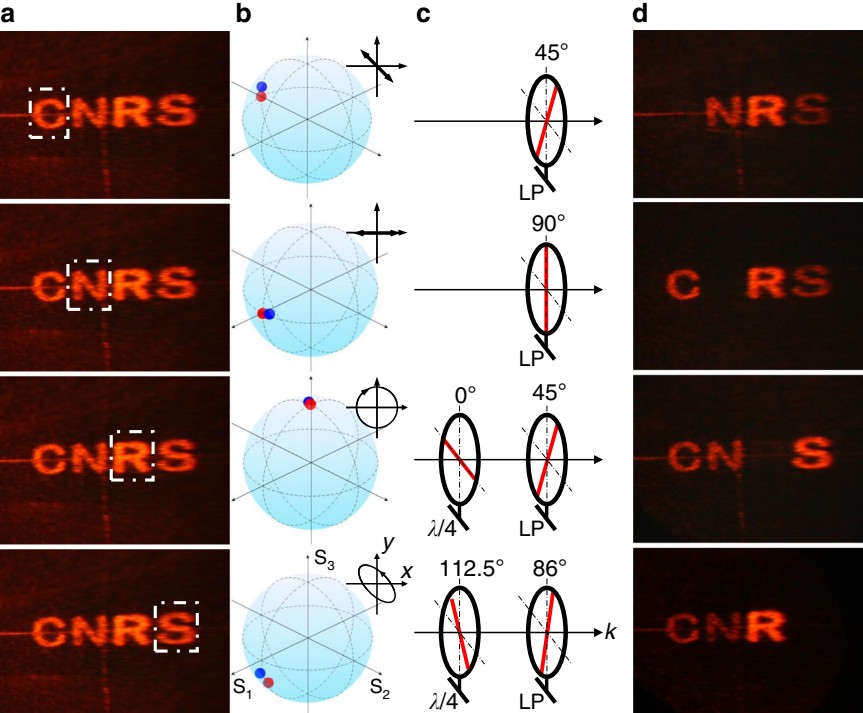

**Fig. 5 Polarization characterization of the meta-hologram. a** Polarimeter position at different letters to analyze the SoP. **b** Measured SoP of the corresponding letters. The red and blue dots indicate the designed and measured polarization, respectively. **c** A linear polarizer with $\theta_{LP} = 45°$ and $\theta_{LP} = 90°$ are put before letter "C" and "N", respectively. A quarter waveplate with $\theta_{\lambda/4} = 0°$ and a linear polarizer with $\theta_{LP} = 45°$ are put before letter "R". A quarter waveplate with $\theta_{\lambda/4} = 112.5°$ and a linear polarizer with $\theta_{LP} = 86°$ are put before letter "S". **d** Photographs of the holographic images indicate that the four letters are blocked correspondently after placing selected polarizer and quarter waveplate.

through the selected quarter waveplate and linear polarizer, the output electric field $\mathbf{E_{out}}$ is described as $\mathbf{E_{out}} = A_{LP}(\theta_{LP})A_{\lambda/4}(\theta_{\lambda/4})\mathbf{E_{in}}$. Selected rotation angle $\theta_{\lambda/4}$ and $\theta_{LP}$ are chosen to block the specific output images (i.e., $\mathbf{E_{out}} = \mathbf{0}$).

**Measurement of efficiency.** The efficiency of the holographic image was defined as the ratio of the total power of the image at the interested order to the incident power. The power of the holographic image was measured by a power meter (Newport: Model 843-R). The incident power was measured as light passing through an aperture with the same size of the meta-hologram.

**Vectorial ptychography.** Ptychography is a recently developed quantitative phase imaging microscopy method inherited from lens-less approaches. In its basic scalar version, is based on the acquisition of series of diffraction intensity patterns recorded by illuminating an object at several overlapping positions with a finite-size spatially-coherent probe (see Supplementary Fig. 9a). If one can assume that the illumination $P$ interacts with the illuminated object $J$ in a multiplicative way, then the diffracted intensity distribution, at the $j$-th scanning position, writes

$$I_j = \left| F\left[J(\mathbf{r}) \times P(\mathbf{r} - \mathbf{r}_j)\right] \right|^2, \quad (4)$$

and one can show that, under proper scanning and sampling measurement conditions, the image of the scanned object (i.e., its complex amplitude) can be reconstructed numerically by means of an iterative algorithm[32], with a transverse resolution mainly determined by the extreme angles with which the diffracted intensity patterns are collected (see Supplementary Fig. 9a).

The vectorial version of ptychography used in this work is a generalized version imaging objects that affects the SoP of light. Namely, the previous equation remains valid, but requires writing the illumination vectorial field distribution $P(r)$ as a so-called Jones vector map[33],

$$P(r) = \begin{bmatrix} P_x(r) \\ P_y(r) \end{bmatrix}, \quad (5)$$

while the object to be reconstructed takes now the mathematical form of Jones matrix map

$$J(r) = \begin{bmatrix} J_{xx}(r) & J_{yx}(r) \\ J_{xy}(r) & J_{yy}(r) \end{bmatrix}. \quad (6)$$

In previous works, we have demonstrated theoretically[34] and experimentally[27] how to use polarized illuminating probes, together with recording of the intensity

diffraction patterns behind an analyzer, in order to reconstruct the Jones matrix map of the investigated object, in addition to the joint reconstruction of the each illumination probe used in the measurement, detailed in ref. [28].

Although there exist various microscopy techniques dedicated to quantitative phase imaging or SoP imaging, vectorial ptychography is the only approach that can image the Jones matrix, which encompasses both aspects together, in a robust and reference-free manner. Here in the context of meta-holograms, the knowledge of the Jones matrix map is crucial since (i) it provides an exhaustive characterization of the component on its entire area at a microscopic resolution, including all amplitude, phase, and polarization effects, (ii) it allows to compute, by simple propagation, the vectorial far field of the holographic image, for any desired input illumination (see Supplementary Fig. 9c).

**Optical vectorial ptychography.** Measurements were carried out on a custom setup[27], operating at the wavelength of 635 nm, which is slightly different to the initial design wavelength of 600 nm due to the setup limitation. However, the polarization of the holographic image is weakly related to the wavelength, only depending on the accumulated phase difference variation between top RCP and bottom LCP beams, however, both vary similarly versus the wavelength[21]. The meta-holograms were placed on a motorized stage (U-780, Physik Instrumente) and scanned under a finite sized probe with effective diameter of 50 μm, selected optically by placing a 2-mm diameter iris diaphragm in the image plane of a ×40 objective lens (ACHN-P, NA 0.65, Olympus). The camera (Stingray F-145B, Allied Vision, 320 × 240 effective pixels of 25.8 × 25.8 μm² after binning) was placed 190 mm after the diaphragm. This configuration results in a final reconstructed map of the Jones matrix spatially sampling at 360 and 490 nm along the horizontal and vertical direction, respectively, which allows to resolve the individual subpixels of the meta-hologram. Consequently to the metasurface operation regime, the subwavelength spacing of the individual nanorods cannot be resolved. The camera acquisition time was set to benefit from the whole dynamical range of the acquisition setup. The raster-scan grid contained 1122 points with average steps of 7.3 μm along the two directions. All scans included additional random step fluctuations of ±50%, in order to avoid periodic reconstruction artefacts. At each scanning position, three linearly polarized probes at angles of 0, 60, 120° in the object plane and three linear analyses at angles 0, 60, 120° were used, resulting in nine different combinations. Object reconstructions were performed by means of the conjugate gradient algorithm, allowing the estimation of Jones matrices of the meta-holograms, jointly with the estimation of the three illumination probes used for the measurement[28]. Jones matrices reconstruction are been obtained after 530 iterations, run on a

multi-graphics-processor-unit (DGX Station, NVIDIA). The vectorial $\mathbf{E(r)}$ exit field is further given, at every point $\mathbf{r}$ of the metasurface, by the matrix product

$$\begin{bmatrix} E_x(\mathbf{r}) \\ E_y(\mathbf{r}) \end{bmatrix} = \begin{bmatrix} J_{xx}(\mathbf{r}) & J_{yx}(\mathbf{r}) \\ J_{xy}(\mathbf{r}) & J_{yy}(\mathbf{r}) \end{bmatrix} \times \begin{bmatrix} \cos(\alpha) \\ \sin(\alpha) \end{bmatrix}, \qquad (7)$$

where the Jones vector to the right represents the LP illumination, with an angle $\alpha$ set to 1.9°, in order to match the orientation of the meta-hologram on the setup. The vectorial far-field $\mathbf{E_f}$ was obtained by computing the far-field propagation by fast Fourier transform (FFT)

$$\begin{bmatrix} E_{fx}(k) \\ E_{fy}(k) \end{bmatrix} = \text{FFT} \begin{bmatrix} E_x(r) \\ E_y(r) \end{bmatrix}, \qquad (8)$$

allowing to quantify, by time evolution, the SoP, ellipticities and polarization orientations for both $\mathbf{E(r)}$ and $\mathbf{E_f}$.

## Data availability
Ptychography raw data and other data that support the findings of this study are available from the corresponding author upon reasonable request.

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

## Acknowledgements
We acknowledge funding from the European Research Council (ERC) under the European Union's Horizon 2020 research and innovation program (Grant agreement nos. 639109 and 724881).

## Author contributions
Q.S. and P.G. conceived the idea and carried out the experiment. Q.S. performed the numerical simulation, calculation of meta-hologram, and optical characterization of meta-hologram; A.B. and P.F. performed the ptychography measurement; Q.S., R.S., P.N., V.B., S.K., and S.C. contributed to the nanofabrication; S.V., B.D., and P.D.M. performed the GaN MBE growth; P.G. supervised and coordinated the project; all the authors contributed and approved the paper.

## Competing interests
The authors declare no competing interests.
