## [Peer Review File · Nature Communications]

Reviewers' comments:

Reviewer #1 (Remarks to the Author):

The manuscript by Song et al. presents a wavefront-shaping, pixelated metasurface that provides arbitrary polarization states when illuminated with linearly-polarized light. Each pixel is able to deflect light at a specific angle with a designed polarization state. The manuscript is well written, the motivation and the goals of the research are clearly stated, and the results are sound and interesting. Previous works have described similar kinds of dielectric metasurfaces able to control light polarization, amplitude and phase in transmissive mode (an example is Ref. 21 of the manuscript - Nature Nanotechnology 10, 937–943 (2015)). However, in the present manuscript there are several elements of novelty that, in my view, significantly expand the functionalities of dielectric metasurfaces for applications in which weight and size are design constraints. In particular, the original aspects introduced in the present work are: (i) the use of non-resonant dielectric elements to make the device response broadband, (ii) the use of a subpixel to realize deflection at specific angles, and (iii) the use of vectorial ptychography which allows extraction of the Jones matrix of each pixel in the metasurface. Overall, the manuscript deserves publication in Nat Comm, although it is my opinion that it may be improved once the authors will clarify some points and provide answers to the following questions:

1. The design criteria of the "meta-molecule" elements are not explained anywhere in the text. Why the GaN "bricks" in the unit cell have the specific shape with large thickness ($1\ \mu\text{m}$) and smaller rectangular cross section? How are those sizes and shape chosen?
2. The "meta-molecule" are referred twice in the text as "antennas". It is my understanding, but I might be wrong, that the purpose of these elements is to introduce birefringence and phase accumulation. In this case, the term "antenna" is not the best choice. In fact, an antenna is traditionally a device that transforms far-fields into near-fields, or the other way around, usually by exploiting resonances in order to make this transformation more effective (better coupling or impedance-matching, etc.). Here the GaN "bricks" seem to act as small, anisotropic lenses. I would use the more generic term meta-molecule or meta-atom in this case, rather than the term antenna.
3. It would be helpful, in order to better understand the physics of the proposed unit cell, to show how the fields (amplitude and/or phase) are localized within the unit cell for each star in Fig.2(a). This would help the reader understanding better the physics at the basis of the proposed metasurface.
4. The role of the periodicity may be clarified. Is there coupling between adjacent unit cells? Does the PB phase somehow depend on the periodicity?
5. The metasurface is fabricated with electron-beam lithography. Is there any chance to realize metasurfaces with similar functionalities by using high-throughput and low-cost alternatives, such as nanoimprint lithography? Is there any possibility to apply the proposed metasurface on curved/conformal substrates?
6. I was not able to find data about the beam spot-size and shape in the experiment. I would make sure that this information is present in the manuscript.
7. Please, check carefully the manuscript for typos and grammar issues. One example: line 21 in the abstract, the sentence "Jones matrix maps that obtained by means of..." should be rephrased.

Reviewer #2 (Remarks to the Author):

This manuscript by Song et. al. present a general method to arbitrarily tailor the phase and polarization with pixelated metasurfaces, which is capable of modulating the wavefront of electromagnetic wave for flat optics applications. As a proof of concept, the authors numerically and experimentally demonstrated holographic imaging with versatile output polarization states based on the proposed method. Overall, I would say this work reads very impressive to me. The topic of research presented is novel, utilization of the ptychography for optical characterization of

full-polarized wavefront shaping metasurfaces seems modern and prospective. The proposed approach is actually practical for the development of meta-devices and systems since the control of phase and polarization are the key roles in optical control. However, there are some technological and physical issues which have to be properly addressed before the paper can be published in Nature Communications. Detailed comments are listed as below:

1. The Eq. (2) described in Page 6 is inconsistent with the equation shown in the inset of Fig. 1d. Based on the generalized Snell's law, I guess the one shown in Page 6 is correct. Please check and correct this.
2. To effectively deflect the normally incident light, a 2π phase shift should be provided within a supercell. However, there are always four unit elements utilized in each pixel shown in Fig. 1c. Since the increment of rotation angle in each pixel is different, I don't expect to see a complete 2π phase shift in every pixel. If so, how does the proposed metasurface effectively deflect the incoming light into desired directions?
3. It seems like the GaN metasurface is designed to operate at the wavelength of 600 nm. However, in Page 14, the authors mentioned that the measurements were carried out on a custom setup operating at the wavelength of 635 nm. Please check and modify it accordingly.
4. I don't find Fig. 7 in the main context. However, the authors mentioned many times about the Fig. 7 in the caption of Fig. 3. Please check and correct them carefully for benefitting readers.
5. In Page 9, the authors mentioned that "...the randomly distributed meta-hologram are shown in Fig. 3h, confirming..." I guess the authors are talking about the Fig. 4h here, please check it.
6. The discussion of Fig. 1d is missing in the main context. In addition, based on the proposed design principle, if now a linearly-polarized light is incident onto the meta-hologram, would we also observe another series of holographic imaging at angles of $-\theta_{tn}$ due to the symmetry of the phase distribution? If yes, I would suggest the authors briefly discussing this point in the supplementary.
7. The scale bar and corresponding scale for Fig. S6b is missing.
8. The optical spectrum and the induced electromagnetic field profile of GaN nano-pillar are absent. To get a deeper understanding of the designed metasurface, I would suggest the authors provide such information in the supplementary.

Reviewer #3 (Remarks to the Author):

In the manuscript, the authors proposed a full-polarization-reconstructed metasurface that can produce arbitrary polarization for wavefront shaping based on PB phase, the function of the four holographic images with different polarization states at different propagation directions has been experimentally demonstrated at the designed wavelength of 600 nm. The demonstrated metasurface is interesting in vectorial holography community and may hold great potential in various applications including the vector beam generation, color display and AR/VR. It might potentially be suitable for publication in Nature Communications, However, there are many issues in this manuscript that have to be properly addressed.

1. The demonstrated concept of converting an input plane wave with linear polarization to a holographic image with arbitrary spatial output polarization. Such functionality was previously demonstrated by Nano Lett. 18(5), 2885-2892 (2018). It is suggested that the authors should make a discussion and comparison.
2. The claim "previous meta-holograms have limiting polarization channels for real applications and are thus unable to generate holographic images with arbitrary polarization from a fixed input polarization" is not proper, because there were already some reports on this recently, such as Nano Lett. 18(5), 2885-2892 (2018), Light Sci. & Appl. 7(1), 95 (2018), ACS Photon. 6(11), 2712-2718 (2019).
3. Although the demonstrated metasurface only based on PB phase, the dimension of the dielectric building block in the bottom and top lines should be adjusted to further control the amplitude difference between RCP and LCP, such that arbitrary elliptical polarization can be generated. As dimension is adjusted, is it still can be broadband?

4. While the paper demonstrated it only at the wavelength of 600 nm, the bandwidth is essential for consideration in application of color display and AR/VR as described by the manuscript. The working bandwidth may be discussed to increase the feasibility of application.
5. The proposed metasurface employ the detour phase induced by the starting orientation angle of both lines in each pixel to address the meta-holograms. However, the detour phase is usually the displacement modulation as demonstrated previously by Nanoscale 8(3), 1588-1594 (2016), Science Advances 2, e1501258 (2016) and Light Sci. & Appl. 7(1), 78 (2018). The connection between the starting orientation angle and displacement modulation should be discussed.
6. The Fig. 1b only plot a Poincare sphere with multiple points. The connection between the sphere and the metasurface design is very weak. The authors may provide the relationship between the designed metasurface parameters and the corresponding generated polarization states to strength it.
7. The holography images are obtained by using the Gerchberg-Saxton algorithm, which leads to that the holographic image has speckles with random phase. Do the speckled image has effect on the polarization modulations using the superposition of left and right circularly polarized beams?
8. In Eq. 2, what is the parameter p ? Is it the period of the single unit-cell or the super-cell? As the super-cell contains at least 4 unit-cells for one holographic image, there definitely exist high-order diffractions. How about efficiency of the meta-hologram?
9. As the diffraction of polarized beams can be analyzed by using the TE and TM components associated with different amplitude and phase, the polarization varies with diffracted angle, for example, a circularly polarized beam will become an elliptically polarized beam at a non-zero diffracted angle. How does the diffracted angle affect the polarization states of the output light from the metasurface?

Reply report to reviewer #1

We are grateful to the Reviewer for the constructive comments and are delighted that the Reviewer recommended the publication of this manuscript. We are happy to address all the reviewers' comments.

Comment: *The design criteria of the “meta-molecule” elements are not explained anywhere in the text. Why the GaN “bricks” in the unit cell have the specific shape with large thickness (1 μm) and smaller rectangular cross section? How are those sizes and shape chosen?*

Answer: As pointed out by the reviewer, the GaN “bricks” serve as a half waveplate to convert the handedness of the circular polarized light. The phase different between the x and y -component should be π , which is accumulated in a large thickness (1 μm) of the dielectric birefringent structures. The relationship of the transmission and phase difference in terms of the length and width of the meta-molecules is given in Supplementary Figure S5 as shown below.

Figure S5. Simulation results of the GaN nano-pillars on sapphire substrate. (a) The top view of a meta-molecule. (b) The transmission phase difference between φ_x and φ_y when the metasurface is impinged by x - and y -polarized light, respectively. (c) The transmission of x -polarized light $|E_x|$. (d) The transmission of y -polarized light $|E_y|$. The star shown in (b-d) represents the selected size for the structure in this paper with $L_x = 230$ nm and $L_y = 120$ nm, where $|E_x|$ and $|E_y|$ are near-unity and $(\varphi_x - \varphi_y)$ is π , so that the structure acts as a perfect half waveplate to convert the circular polarized light.

Comment: *The “meta-molecule” are referred twice in the text as “antennas”. It is my understanding, but I might be wrong, that the purpose of these elements is to introduce birefringence and phase accumulation. In this case, the term “antenna” is not the best choice. In fact, an antenna is traditionally a device that transforms far-fields into near-fields, or the other way around, usually by exploiting resonances in order to make this transformation more effective (better coupling or impedance-matching, etc.). Here the GaN “bricks” seem to act as small, anisotropic lenses. I would use the more generic term meta-molecule or meta-atom in this case, rather than the term antenna.*

Answer: As suggested by the reviewer, the term “antenna” is replaced by “meta-molecule” in the revised manuscript.

Comment: It would be helpful, in order to better understand the physics of the proposed unit cell, to show how the fields (amplitude and/or phase) are localized within the unit cell for each star in Fig.2(a). This would help the reader understanding better the physics at the basis of the proposed metasurface.

Answer: As suggested by the reviewer, the magnetic field distribution of one meta-molecule is provided in Supplementary Figure S6 as shown below.

Figure S6. Simulated results of the magnetic field distribution (a) H_x and (b) H_y with the illumination of y - and x -polarized light, respectively. (c) H_x and H_y distribution along z -direction at $y = 0$. It can be observed that there are 5 and 5.5 oscillations for H_x and H_y in GaN nano-pillars, respectively, leading to a π phase difference between H_x and H_y . (d) H_y distribution along x -direction at $z = 185$ nm. It is seen that the magnetic field is strongly confined in the GaN nano-pillars, resulting in a relatively weak coupling between adjacent meta-molecules.

Comment: The role of the periodicity may be clarified. Is there coupling between adjacent unit cells? Does the PB phase somehow depend on the periodicity?

Answer: The periodicity is chosen to be subwavelength to avoid unwanted diffraction grating, but not too small due to the fabrication limitation. The coupling between adjacent unit cells is very weak, since all the electromagnetic field is confined in the GaN nano-pillars as shown in Supplementary Figure S6(d). The discussion of the coupling is added in the caption of Figure S6(d) as, “(d) H_y distribution along x -direction at $z = 185$ nm. It is seen that the magnetic field is strongly confined in the GaN nano-pillars, resulting in a relatively weak coupling between adjacent meta-molecules.” and more discussion about the period is added in the revised manuscript as, “The period of the meta-molecule is 300 nm, which is subwavelength to avoid unwanted diffraction grating, while remaining moderate to achieve fabrication.” in Line 122 Page 7.

The PB phase is mainly related to the orientation of the GaN nano-pillars. The periodicity will affect the polarization conversion efficiency of the circular polarized light as shown in Supplementary Fig. S7(b). The size of the GaN nano-pillars is kept at $L_x = 230$ nm and $L_y = 120$ nm. The conversion efficiency is higher than 90% when the period is between 275 nm and 340 nm. It drops fast when the period is larger than 350 nm due to the opening of a

diffractive channel in the substrate. In this paper, we choose the period of 300 nm for the design. The figure and caption of Figure S7(b) is added as below.

Figure S7. Simulated results of the polarization conversion efficiency from LCP to RCP when the geometric size of GaN nano-pillars is chosen as $L_x = 230$ nm and $L_y = 120$ nm, ... (b) when the period is changed from 250 nm to 450 nm with incident wavelength of $\lambda = 600$ nm. The conversion efficiency is higher than 90% when the period is between 275 nm and 340 nm. It drops fast when the period is larger than 350 nm due to diffraction in the substrate. We choose $p = 300$ nm in this paper.

Comment: *The metasurface is fabricated with electron-beam lithography. Is there any chance to realize metasurfaces with similar functionalities by using high-throughput and low-cost alternatives, such as nanoimprint lithography? Is there any possibility to apply the proposed metasurface on curved/conformal substrates?*

Answer: As pointed out by the reviewer, there is possibility of realizing metasurface with similar functionalities by nanoimprint lithography using high dielectric constant and low loss materials. The proposed metasurface can also be applied to the curved/conformal substrates by using soft materials like PDMS demonstrated in Ref 29-31. The discussion is added in the Methods as, “It is worth to note that there is possibility of realizing metasurface with similar functionalities by nanoimprint lithography using high dielectric constant and low loss materials. The proposed metasurface can be further applied to the curved or conformal substrates by using soft materials like PDMS²⁹⁻³¹.”

Comment: *I was not able to find data about the beam spot-size and shape in the experiment. I would make sure that this information is presented in the manuscript.*

Answer: As suggested by the reviewer, the details of the beam spot-size and shape have been added in the caption of Figure S8 as, “...A laser beam with diameter of 3 mm at the wavelength of 600 nm propagates through a linear polarizer and lens to weakly focuses on the meta-hologram...” in the Supplementary.

Comment: *Please, check carefully the manuscript for typos and grammar issues. One example: line 21 in the abstract, the sentence “Jones matrix maps that obtained by means of...” should be rephrased.*

Answer: As suggested by the reviewer, the sentence is rephrased as “Vectorial ptychography technique is introduced for mapping the Jones matrix to monitor...”. The typos and grammar issues in this paper have been carefully checked.

In summary, we have addressed all the comments of reviewer #1.

Reply report to reviewer #2

We are grateful to the Reviewer for the constructive comments and are delighted that the Reviewer recommended the publication of this manuscript. We are happy to address all the comments by the Reviewer.

Comment: The Eq. (2) described in Page 6 is inconsistent with the equation shown in the inset of Fig. 1d. Based on the generalized Snell's law, I guess the one shown in Page 6 is correct. Please check and correct this.

Answer: As pointed out by the reviewer, we have corrected the typo in Fig. 1d accordingly.

Comment: To effectively deflect the normally incident light, a 2π phase shift should be provided within a supercell. However, there are always four unit elements utilized in each pixel shown in Fig. 1c. Since the increment of rotation angle in each pixel is different, I don't expect to see a complete 2π phase shift in every pixel. If so, how does the proposed metasurface effectively deflect the incoming light into desired directions?

Answer: As pointed out by the reviewer, the increment of rotation angle in each pixel is designed to control the deflected angle of the holographic images. A complete 2π phase shift (i.e. the increment of rotation angle larger than 90°) can be achieved when the designed deflected angle is higher than 30° according to Eq. 2. The number of elements in each pixel is discussed in Supplementary Fig. S5 as, “**Figure S4. Determination of the number of meta-molecules within one supercell.** (a) Assume that there are N point light sources with phase increment of $-\varphi$, the electric field scattered from each source is $e^{-ij\varphi}$ ($j = 1, 2, \dots, N$). Therefore, the far field pattern \vec{E}_f can be described as the sum of all sources, i.e., $\vec{E}_f = \sum_{j=1}^N e^{-ij\varphi} e^{ik(|\vec{r}_j| - |\vec{r}|)} = \sum_{j=1}^N e^{-ij\varphi} e^{ik(\frac{N+1}{2} - j)p \sin \omega}$, where k is the wavenumber in free space, p is the period of the array, $|\vec{r}|$ is the distance between the origin and the far field in the direction angle of ω , $|\vec{r}_j|$ is the distance between the j th source and the far field in the direction angle of ω . A normalized electric field pattern \vec{E}_{fn} by dividing \vec{E}_f by N is used to compare the intensity profile with different N . Considering the working wavelength of $\lambda = 600 \text{ nm}$, period of $p = 300 \text{ nm}$, wavenumber of $k = 2\pi/\lambda$, the normalized far field pattern \vec{E}_{fn} can be described as $\vec{E}_{fn} = \frac{\sum_{j=1}^N e^{-ij\varphi} e^{i\pi(\frac{N+1}{2} - j) \sin \omega}}{N}$, $N = 2, 3, 4, \dots$ (b) The normalized far field pattern with different numbers of sources N . The phase differences between each source is chosen as $\varphi = 60^\circ$. It is shown that when N is changed from 2 to 6 (i.e., the phase difference within a supercell is changed from 120° to 360°), the radiation angle always keeps at 19.5° , while the intensity radiated to the interested angle is enhanced. However, when N becomes larger, the size of the pixel is increasing, which decreases the total number of the pixels. As a trade-off, we choose $N = 4$ to design the pixels.”

Figure S4. Determination of the number of meta-molecules within one supercell.

Comment: *It seems like the GaN metasurface is designed to operate at the wavelength of 600 nm. However, in Page 14, the authors mentioned that the measurements were carried out on a custom setup operating at the wavelength of 635 nm. Please check and modify it accordingly.*

Answer: As pointed out by the reviewer, the ptychography setup is characterized at the wavelength of 635 nm due to the constraint of the setup. However, the polarization of the proposed meta-hologram is fixed after defining the orientation angle and geometric size of the structures ($\psi = \delta$ and $\chi = \frac{1}{2} \arcsin \frac{a_R^2 - a_L^2}{a_R^2 + a_L^2}$), which is not related to the wavelength. Therefore, the polarization characterization of Fig. 4h is not affected by the chosen wavelength. The discussion of the difference of the wavelength is added in the revised Methods as, “Measurement were carried out on a custom setup, operating at the wavelength of 635 nm, which is slightly different to the initial design wavelength of 600 nm due to the setup limitation. However, the polarization of the holographic image is weakly related to the wavelength, only depending on the accumulated phase difference variation between top RCP and bottom LCP beams, however both vary similarly versus the wavelength²².” in Line 277 Page 14.

Comment: *I don't find Fig. 7 in the main context. However, the authors mentioned many times about the Fig. 7 in the caption of Fig. 3. Please check and correct them carefully for benefitting readers.*

Answer: We thank the referee for the careful reading of the manuscript and for pointing out our mistake. The caption of Fig. 3 is revised as, “Uniformly and randomly distributed sub-pixels. (a) – (d) The sub-pixels in one super-pixel are distributed with four different arrangement. (e) A 100×100 matrix with randomly distributed integer of 1, 2, 3, and 4 is generated using MATLAB. Each numeral is corresponding to the sub-pixels Fig. 3a-3d. (f) A beam deflector is designed with sub-pixels uniformly distributed using configurations shown in Fig. 3a. The four sub-pixels C1, C2, C3 and C4 are designed with interested order at 15° , 18.5° , 22° and 25.5° , respectively. (g) Measured far-field intensity pattern shows that a series of grating orders β_m^{cn} ($m = \pm 1$, $n = 1, 2, 3, 4$) is generated. (h) Measured angles of the grating orders agree well to the simulated results. (i) A beam deflector design with randomly distributed sub-pixels based on the random matrix in Fig. 3e. (j) Measured far-field intensity pattern shows the grating order is destroyed due to a variable lattice constant d . (k) Only interested order is obtained without other grating orders.”

Comment: *In Page 9, the authors mentioned that “...the randomly distributed meta-hologram are shown in Fig. 3h, confirming...” I guess the authors are talking about the Fig. 4h here, please check it.*

Answer: As pointed out by the reviewer, the sentence has been corrected as “...the randomly distributed meta-holograms are shown in Fig. 4h, confirming...” in Line 182 Page 9.

Comment: *The discussion of Fig. 1d is missing in the main context. In addition, based on the proposed design principle, if now a linearly-polarized light is incident onto the meta-hologram, would we also observe another series of holographic imaging at angles of $-\theta_{in}$ due to the symmetry of the phase distribution? If yes, I would suggest the authors briefly discussing this point in the supplementary.*

Answer: As pointed out by the reviewer, the discussion of Fig. 1d has been added in the revised manuscript as, “The SoP of the four images can be independently designed to realize a full-polarization-reconstructed and multi-directional meta-hologram as shown in Fig. 1d.” in Line 114 Page 6. Indeed, there are symmetric holographic images at the diffraction angle of -

θ_i with polarization of $|n_{-\theta}\rangle = a_R|L\rangle + a_L e^{i2\delta}|R\rangle$. The discussion has been added in the caption of Supplementary Fig. S10 as “...Another diffraction order presenting identical intensity distribution at the opposite angle of $-\theta_i$ is obtained with polarization of $|n_{-\theta}\rangle = a_R|L\rangle + a_L e^{i2\delta}|R\rangle$, which for simplicity is not presented in this paper.”.

Comment: The scale bar and corresponding scale for Fig. S6b is missing.

Answer: As pointed out by the reviewer, the scale bar is added in Supplementary Fig. S10b and the corresponding scale is included in the caption of Fig. S10b as, “The red scale bar is $1\mu\text{m}$ ”.

Comment: The optical spectrum and the induced electromagnetic field profile of GaN nano-pillar are absent. To get a deeper understanding of the designed metasurface, I would suggest the authors provide such information in the supplementary.

Answer: As pointed out by the reviewer, the optical spectrum of GaN nano-pillar is included in Supplementary Fig. S7(a) as shown below.

Figure S7. Simulated results of the polarization conversion efficiency from LCP to RCP when the geometric size of GaN nano-pillars is chosen as $L = 230\text{ nm}$ and $w = 120\text{ nm}$, (a) when the wavelength of the incident light is changed from 450 nm to 850 nm with $p = 300\text{ nm}$.

And the induced electromagnetic field profile of GaN nano-pillar is shown in Supplementary Fig. S6 as below.

Figure S6. Simulated results of the magnetic field distribution (a) H_x and (b) H_y with the illumination of y - and x -polarized light, respectively. (c) H_x and H_y distribution along z -direction at $y = 0$. It can be observed that there are 5 and 5.5 oscillations for H_x and H_y in GaN structure, respectively, leading to a π phase difference between H_x and H_y . (d) H_y distribution along x -direction at $z = 185$ nm. It is seen that the magnetic field is strongly confined in the GaN nano-pillars, resulting in a relatively weak coupling between adjacent meta-molecules.

In summary, we have addressed all the comments of reviewer #2.

Reply report to reviewer #3

We are grateful to the Reviewer for the constructive comments and are delighted that the Reviewer recommended the potential of publication of this manuscript. We are happy to address all the comments by the Reviewer.

Comments: *Such functionality was previously demonstrated by Nano Lett. 18(5), 2885-2892 (2018). It is suggested that the authors should make a discussion and comparison.*

And

The claim “previous meta-holograms have limiting polarization channels for real applications and are thus unable to generate holographic images with arbitrary polarization from a fixed input polarization” is not proper, because there were already some reports on this recently, such as Nano Lett. 18(5), 2885-2892 (2018), Light Sci. & Appl. 7(1), 95 (2018), ACS Photon. 6(11), 2712-2718 (2019).

Answer: We thank the referee for this suggestion which has been included and discussed in the main text as follow: “Among the other attempts in achieving polarization dependent holographic metasurfaces, we point out that reflective plasmonic metasurfaces utilizing two plasmonic nanorods per period with different distance and orientation angle can handle arbitrary polarization states but only efficient for oblique incidence.¹⁸ Vectorial meta-holograms are also realized by combining both the propagation phase and PB phase, which suffers from narrow bandwidth^{19,20}.” in Line 47 Page 3.

Comment: *Although the demonstrated metasurface only based on PB phase, the dimension of the dielectric building block in the bottom and top lines should be adjusted to further control the amplitude difference between RCP and LCP, such that arbitrary elliptical polarization can be generated. As dimension is adjusted, is it still can be broadband?*

Answer: As pointed out by the reviewer, the proposed metasurface is broadband for all the output linear polarizations and a few elliptical polarizations where the dimension of the building blocks is uniform. It becomes narrow band when the dimension of the building block in the bottom and top lines are not the same. The discussion is added in the revised manuscript as, “In addition, the proposed method is broadband for all of the output linear polarizations.” in Line 64 Page 4 and “Since the dimension of the building blocks are uniform for phase-only superposition method, the metasurface is broadband for all of the output linear polarizations. It becomes narrow band when the dimension of the building block in the bottom and top lines are not the same.” in the caption of Supplementary Fig. S1.

Comment: *While the paper demonstrated it only at the wavelength of 600 nm, the bandwidth is essential for consideration in application of color display and AR/VR as described by the manuscript. The working bandwidth may be discussed to increase the feasibility of application.*

Answer: As suggested by the reviewer, we characterize the holographic images at visible range from 475 nm to 675 nm in Supplementary Fig. S11(a-i) and the corresponding SoP is given in Supplementary Fig. S11(j). The figure and caption of Fig. S11 are added as shown below.

Figure S11. Measurement results in a wide bandwidth. (a-i) The holographic images of “alive Schrödinger’s cat” representing the SoP of LP-45° with different incident wavelength from 475 nm to 675 nm. (j) The corresponding SoP of the images on Poincaré sphere agrees well to the designed SoP of LP-45°, exhibiting broadband feature for output linear polarization.

Comment: The proposed metasurface employ the detour phase induced by the starting orientation angle of both lines in each pixel to address the meta-holograms. However, the detour phase is usually the displacement modulation as demonstrated previously by *Nanoscale* 8(3), 1588-1594 (2016), *Science Advances* 2, e1501258 (2016) and *Light Sci. & Appl.* 7(1), 78 (2018). The connection between the starting orientation angle and displacement modulation should be discussed.

Answer: As pointed out by the reviewer, the detour phase is discussed in the revised manuscript as, “The starting orientation angle of both lines in each pixel results in a detour phase that is exploited to address phase information of the meta-holograms (see Supplementary Fig.2 and Fig. 3), which introduces another freedom to control the detour phase without shifting the structures by displacement of the aperture²³⁻²⁵.” in Line 103 Page 6.

Comment: The Fig. 1b only plot a Poincare sphere with multiple points. The connection between the sphere and the metasurface design is very weak. The authors may provide the relationship between the designed metasurface parameters and the corresponding generated polarization states to strength it.

Answer: As suggested by the reviewer, the discussion of the Poincaré sphere and the metasurface design is added in the revised manuscript as, “The azimuth angle ψ and ellipticity angle χ of $|n\rangle$ are derived as $\psi = \delta$ and $\chi = \frac{1}{2} \arcsin \frac{a_R^2 - a_L^2}{a_R^2 + a_L^2}$. In this way, an arbitrary full-polarization can be reconstructed by changing δ from $-\pi$ to π and a_L, a_R from 0 to 1, covering the entire Poincaré sphere as shown in Fig. 1b. The changing of δ leads to the shifting of the SoP along the latitude of the Poincaré sphere, while the changing of a_R and a_L makes the SoP shift along the longitude direction. When $\frac{a_R}{a_L}$ is increasing, the SoP moves towards the top of the Poincaré sphere and vice versa.” in Line 94 Page 5.

Figure 1. Full-polarization-reconstruction and multi-directional meta-hologram... (b) Plotted Poincaré sphere covered full-polarization with phase difference α from $-\pi$ to π and amplitude a_R and a_L from 0 to 1.

Comment: *The holography images are obtained by using the Gerchberg-Saxton algorithm, which leads to that the holographic image has speckles with random phase. Do the speckled image has effect on the polarization modulations using the superposition of left and right circularly polarized beams?*

Answer: As pointed out by the reviewer, it is true that the holographic image generated by GS algorithm leads to speckles with random phase. However, the speckled image doesn't affect the output polarization which is only determined by the pixelated deflector itself rather than the interference between different pixels. The discussion is added in the revised manuscript as, "Although the Gerchber-Saxton algorithm creates speckled images with random phase, it does not affect the SoP of the meta-hologram which is determined at the pixel level, instead of relying on the interference between different pixels." in Line 151 Page 8. Note that within each pixel, the phase difference between L and R remains fixed by design thus produce a fixed output polarization.

Comment: *In Eq. 2, what is the parameter p ? Is it the period of the single unit-cell or the super-cell? As the super-cell contains at least 4 unit-cells for one holographic image, there definitely exist high-order diffractions. How about efficiency of the meta-hologram?*

Answer: As pointed out by the reviewer, the parameter p is the period of one single unit-cell as shown in the inset picture of Fig. 2a. Indeed, there are high-order diffractions induced in the holographic image. When the interested angle is targeted at 30° , a high order image induced by the diffraction is shown near the central spot as shown in Fig. 2e. However, the intensity of the high order images is very low compared to the interested order. Meanwhile, this high order images can be eliminated by randomly distributing the pixels to avoid the grating effect. The discussion is added in the revised manuscript as, "The absolute efficiency of each image at the interested order is $\eta_{LP+45^\circ} = 17.2\%$, $\eta_{LP-H} = 15.3\%$, $\eta_{RCP} = 25.4\%$ and $\eta_{EP} = 20.3\%$, respectively. The drawback with this encoding pixelation method is that residual ghost images are produced due to the remaining polarization periodicity, equivalent to a grating effect according to (see Supplementary Fig. 7).

$$\beta_m = \arcsin\left(\sin \theta_t + \frac{2m\pi}{k_0 d}\right) \quad (3)$$

Indeed, grating effect occurs for uniformly distributed sub-pixel arrangement, producing multiple grating orders. To better comprehend the encoding method, we have proposed to demonstrate simple beam deflectors integrated with four sub-pixels

arrays using two types of arrangement: uniformly or randomly distributed sub-pixels as shown in Fig. 3. The results show that for randomly distributed sub-pixels, i.e. when the lattice constant d in Eq. 3 becomes variable, the grating effect is cancelled. As a result, a pure desired order is obtained as shown in Fig. 3j.” in Line 157 Page 8.

Comment: *As the diffraction of polarized beams can be analyzed by using the TE and TM components associated with different amplitude and phase, the polarization varies with diffracted angle, for example, a circularly polarized beam will become an elliptically polarized beam at a non-zero diffracted angle. How does the diffracted angle affect the polarization states of the output light from the metasurface?*

Answer: As pointed out by the reviewer, the diffracted angle doesn't affect the polarization states of the output light from the metasurface. The metasurface converts the handedness of the input circular polarized beam and imposes a PB phase gradient in the converted circular polarized beam, thus that only the converted circular polarized beam can be deflected to the interested angle. Therefore, a pure circular polarized beam is obtained instead of changing to an elliptically polarized beam. The discussion of the polarization is added in the revised manuscript as, “It should be noted that the output polarization from top and bottom lines to the interested deflected angle is pure RCP and LCP. Since the metasurface converts the handedness of the input CP beam and imposes a PB phase gradient in the converted CP beam, only the converted CP beam can be deflected to the interested angle and thus that pure CP beams from top and bottom supercells are obtained.” in Line 81 Page 5.

In summary, we have addressed all the comments of reviewer #3.

REVIEWERS' COMMENTS

Reviewer #1 (Remarks to the Author):

The point-by-point authors' response is very convincing and, in my view, properly addresses the issues raised in the review reports. I think this work is very interesting, and that the manuscript has been improved and it deserves publication.

Reviewer #2 (Remarks to the Author):

The authors properly response all my questions and comments. Now I can agree this manuscript to be accepted for publication with current version.

Reviewer #3 (Remarks to the Author):

The authors addressed all the points of reviewer's, publication is recommended.